# A *COL7A1* Variant in a Litter of Neonatal Basset Hounds with Dystrophic Epidermolysis Bullosa

**DOI:** 10.3390/genes11121458

**Published:** 2020-12-04

**Authors:** Teresa Maria Garcia, Sarah Kiener, Vidhya Jagannathan, Duncan S. Russell, Tosso Leeb

**Affiliations:** 1Department of Biomedical Sciences, Carlson College of Veterinary Medicine, Oregon State University, Corvallis, OR 97331, USA; teresa.garcia@oregonstate.edu; 2Institute of Genetics, Vetsuisse Faculty, University of Bern, 3001 Bern, Switzerland; sarah.kiener@vetsuisse.unibe.ch (S.K.); vidhya.jagannathan@vetsuisse.unibe.ch (V.J.); 3Dermfocus, University of Bern, 3001 Bern, Switzerland

**Keywords:** *Canis lupus familiaris*, whole genome sequence, skin, dermatology, genodermatosis, collagen VII, precision medicine

## Abstract

We investigated three neonatal Basset Hound littermates with lesions consistent with epidermolysis bullosa (EB), a group of genetic blistering diseases. A clinically normal bitch was bred to her grandfather by artificial insemination. Out of a litter of seven puppies, two affected puppies died and one was euthanized, with these puppies being submitted for diagnostic necropsy. All had multiple bullae and ulcers involving the nasal planum and paw pads, as well as sloughing claws; one puppy also had oral and esophageal ulcers. The complete genome of one affected puppy was sequenced, and 37 known EB candidate genes were assessed. We found a candidate causative variant in *COL7A1*, which encodes the collagen VII alpha 1 chain. The variant is a complex rearrangement involving duplication of a 107 bp region harboring a frameshift deletion of 7 bp. The variant is predicted to truncate more than 75% of the open reading frame, p.(Val677Serfs*11). Targeted genotyping of this duplication confirmed that all three affected puppies were homozygous for the duplication, whereas 12 unaffected Basset Hounds did not carry the duplication. This variant was also not seen in the genomes of more than 600 dogs of other breeds. *COL7A1* variants have been identified in humans and dogs with dystrophic epidermolysis bullosa (DEB). The identified *COL7A1* variant therefore most likely represents the causative variant and allows the refinement of the preliminary EB diagnosis to DEB.

## 1. Introduction

Epidermolysis bullosa (EB) is a heterogeneous group of genetic disorders associated with skin fragility and blistering, as well as secondary to minor mechanical trauma [1]. Four categories, or “classical types”, of EB have been described in humans, broadly distinguished by the location of blistering in the skin [1]. Epidermolysis bullosa simplex (EBS) causes blistering in basal or suprabasal keratinocytes; junctional epidermolysis bullosa (JEB) causes blistering at the lamina lucida of the basement membrane zone; dystrophic epidermolysis bullosa (DEB) causes blistering in the superficial dermis; and, in Kindler epidermolysis bullosa (KEB), lesions are present in multiple levels of the basement membrane zone, and affected individuals have clinical features not seen in the other types [2].

Further sub-classification in humans describes the phenotype, mode of inheritance, targeted protein and relative expression, the gene involved, and the genetic variant present, with variant specifics if available [1,2]. A recent reclassification of human DEB categorizes phenotypes according to recessive, dominant, and compound heterozygous allelic variation, however, certain clinical presentations are associated with more than one pattern of inheritance [1]. Thus far, all types of DEB in humans have been attributed to variants of the *COL7A1* gene, which codes for the collagen VII alpha 1 chain, although a recent case describes a DEB-like phenotype found in an individual with a variant in *PLOD3*, which encodes for lysyl-hydroxylase 3 [1,3,4]. A human DEB variant database has documented 659 variants of *COL7A1* [5]. Recessive forms of DEB may be more severe than dominant types, but there is considerable phenotypic overlap between the two [1,6,7,8].

In dogs, cases of putative EBS, JEB, and DEB have been described, although fewer subtypes and genetic variants than in humans have been recognized [9,10,11,12,13,14,15,16,17,18]. A recent case report of recessive DEB in two neonatal Central Asian Shepherd siblings found a homozygous nonsense variant of *COL7A1* that resulted in a premature stop codon, likely preventing the production of functional protein [14] (OMIA 000341-9615). A mild, recessive form of DEB in a family of Golden Retrievers was identified, and genetic analysis found a homozygous missense variant, p.Gly1906Ser, in *COL7A1* [15,16]. Another report of putative DEB in an Akita Inu was diagnosed via observation of the periodic acid-Schiff (PAS)-positive basement membrane at the roof of the blister, separation of the tissues beneath the lamina densa, and decreased anchoring fibrils on electron microscopy. The causative genetic variant was not reported [17]. Herein, we describe cases of DEB in a single litter of neonatal Basset Hounds, a breed in which, to our knowledge, this condition has not yet been reported.

## 2. Materials and Methods

### 2.1. Ethics Statement

All animal experiments were performed according to local regulations. Puppies in this study were privately owned and deceased animals were submitted for diagnostic necropsy. The Cantonal Committee for Animal Experiments approved the collection of blood samples for the 12 normal Basset Hounds (Canton of Bern; permit 75/16).

### 2.2. Animal Selection

A clinically normal, approximately two-and-a-half-year-old Basset Hound bitch was bred to her maternal grandfather. He had been deceased for over 20 years, and the veterinarian reported that the last of 5 straws of frozen semen were used for this breeding.

An unrelated, stillborn, neonatal puppy with grossly normal skin was the control for electron microscopy. Control blood samples were from 12 clinically normal Basset Hounds in the Vetsuisse Biobank.

### 2.3. Necropsy, Histopathology, and Electron Microscopy

All pathological specimens were evaluated by a board-certified veterinary pathologist (D.S.R.) and a veterinary pathologist in training (T.M.G.). A full suite of tissues from all 3, and skin from the control puppy, were immersed in 10% neutral buffered formalin and processed for routine histopathology. Skin sections were routinely stained with hematoxylin and eosin, gram, and periodic acid–Schiff. A smaller subset of tissues were saved from all three affected puppies and were stored at −20 °C for DNA extraction. For transmission electron microscopy (TEM), sections of skin from the euthanized puppy were collected and immediately suspended in 3% glutaraldehyde. Skin from the normal puppy was suspended in modified Karvosky fixative. Samples were rinsed with 0.1 M sodium cacodylate buffer, and post-fixed in 1.5% potassium ferrocyanide and 2% osmium tetroxide in water. Samples were stained with T-O-T-O, followed by lead aspartate, and were dehydrated in a graded series of acetone (10%, 30%, 50%, 70%, 90%, 95%, 100%) for 10–15 min each. They were then infiltrated with araldite resin, ultrathin sectioned, and placed on TEM grids. Images were taken in a FEI Helios Nanolab 650 Scanning Electron Microscope in STEM mode.

### 2.4. DNA Extraction

Genomic DNA was either isolated from frozen liver tissue with a Maxwell RSC Tissue DNA Kit or from ETDA blood samples with the Maxwell RSC Whole Blood Kit using a Maxwell RSC instrument (Promega, Dübendorf, Switzerland).

### 2.5. Whole Genome Sequencing

An Illumina TruSeq PCR-free DNA library with ≈400 bp insert size of an affected Basset Hound puppy was prepared. We collected 263 million 2 × 150 bp paired-end reads on a NovaSeq 6000 instrument (30× coverage). The reads were mapped to the dog reference genome assembly CanFam3.1 as previously described [19]. The sequence data were deposited under the study accession PRJEB16012 and the sample accession SAMEA6862953 at the European Nucleotide Archive.

### 2.6. Variant Calling

Variant calling was performed using GATK HaplotypeCaller [20] in gVCF mode as described [19]. To predict the functional effects of the called variants, we used SnpEff [21] software together with NCBI annotation release 105 for the CanFam3.1 genome reference assembly. For variant filtering, we used 73 control genomes (Appendix A). Four of the 73 control genomes were derived from Basset Hounds. The same Basset Hounds were also included in the 12 Basset Hound controls for targeted genotyping. Numbering within the canine *COL7A1* gene corresponds to the NCBI RefSeq accession numbers NM_001002980.1 (mRNA) and NP_001002980.1 (protein).

### 2.7. Sanger Sequencing and Genotyping

The *COL7A1*:[c.2028_2034;c.1993_2050+56dup] variant was genotyped by direct Sanger sequencing of PCR amplicons (Appendix A). A 389 bp (or 489 bp in case of the mutant allele) PCR product was amplified from genomic DNA using AmpliTaqGold360Mastermix (Thermo Fisher Scientific, Waltham, MA, USA) together with primers 5‘-GTG GGA GGG CTA TAG GGA AG-3‘ (Primer F) and 5′-AAA GGA GGC CAA AGG AGA AA-3′ (Primer R). After treatment with exonuclease I and alkaline phosphatase, amplicons were sequenced on an ABI 3730 DNA Analyzer (Thermo Fisher Scientific, Waltham, MA, USA). Sanger sequences were analyzed using the Sequencher 5.1 software (GeneCodes, Ann Arbor, MI, USA). A 5200 Fragment Analyzer was used for the sizing and visualization of PCR products (Agilent, Santa Clara, CA, USA).

## 3. Results

### 3.1. Clinical History and Necropsy Findings

Out of seven puppies, one was stillborn with an abdominal wall defect and protruding intestines; another died at 12 h of age following weakness and dyspnea. Of the remaining puppies, two were born with erosions of the skin on the muzzle. Within two days after birth, these two and one additional puppy were noted with blisters on the paw pads, multiple sloughed nails, and crusts around the ear canals. Two such affected puppies died naturally during the night, between the second and third day of life, were frozen near the time of death, and submitted for routine necropsy approximately 3 months later. The third was humanely euthanized between 48–72 h after birth and submitted for necropsy approximately 1 week after death. The two remaining puppies of the litter were clinically normal at three months of age.

Grossly, the euthanized puppy had bullae up to 8 mm in diameter affecting the nasal planum, dorsal muzzle, and paw pads (digital, metacarpal, and metatarsal) from all four limbs (Figure 1). Several nails were sloughed. There were multifocal to coalescing areas of ulcerations on the tongue and lips. The other two affected puppies had similar gross lesions, including bullae and ulceration of the nasal planum and dorsal muzzle, and of the paw pads. Both had multiple sloughed claws. The gross diagnosis was multifocal, cutaneous, and mucocutaneous bullous dermatopathy with ulceration, consistent with a severe form of EB.

### 3.2. Histologic Findings

All puppies had similar microscopic changes. Most prominently, all three had sub-basilar bullae in the paw pads and nasal planum (Figure 1). In adjacent, less affected areas, there was clefting at the dermal–epidermal junction with accumulation of proteinaceous fluid. Sections stained with PAS found positively stained material at both the roof and floor of the bullae. Ruptured bullae were associated with granulation tissue, lymphoplasmacytic and neutrophilic inflammation, and many surface-associated Gram-positive and Gram-negative bacteria. The euthanized puppy also had severe lingual and esophageal ulceration, with granulation tissue and surface-associated bacteria. All other organs were microscopically normal.

### 3.3. Electron Microscopy

While many structures of the basement membrane zone were identified, inadequate preservation (autolysis) precluded identification of the specific location of clefting, as well as abnormalities within structures such as hemidesmosomes and anchoring fibrils. Images are available in the Appendix A (Appendix A).

### 3.4. Genetic Analysis

We sequenced the genome of one affected puppy at 30× coverage and searched for private homozygous and heterozygous variants in 37 known candidate genes (Appendix A) that were not present in the genome sequences of 73 control dogs (Table 1, Appendix A).

The automated analysis identified three homozygous protein-changing variants in *COL7A1*, a known candidate for DEB in humans and dogs. The GATK software called three independent smaller variants within a stretch of 11 nucleotides. However, visual inspection of the short-read alignments revealed that these three variants were not correctly called (Figure 2). They are part of a single complex duplication event spanning parts of exon 15 and intron 15 of the *COL7A1* gene, starting at position 40,524,267 and ending at 40,524,380 on chromosome 20 (CanFam3.1 assembly). The variant can be designated as NM_001002980.1:[c.2028_2034del; c.1993_2050+56dup]. The variant causes a frameshift. Assuming the formation of a transcript, in which the mutant exon 15 is spliced to the unchanged exon 16 without inclusion of duplicated sequences, this frameshift is predicted to truncate 2260 amino acids (76%) from the C-terminus of the wildtype COL7A1 protein NP_01002980.1:p.(Val677Serfs*11). We did not investigate whether any mutant protein is expressed or whether the complex rearrangement leads to nonsense mediated decay or altered splicing of the mutant transcript.

We confirmed the exact sequence of the variant in *COL7A1* by Sanger sequencing in one of the affected puppies (Appendix A) and genotyped all three affected puppies and an additional 12 control Basset Hounds for this variant. All three affected puppies were homozygous for the mutant allele, while none of the 12 control Basset Hounds carried this allele. The duplication was also absent from the 73 control genomes used in variant filtering and another 588 publicly available genomes from other dog breeds and wolves reported in the study by [19].

## 4. Discussion

A bullous dermatopathy was diagnosed in three neonatal Basset Hound puppies from a single litter. Given the constellation of gross and microscopic lesions, age of onset, and multiple instances of inbreeding in the pedigree, EB was our top differential diagnosis. Our finding of a homozygous, candidate-causative variant in *COL7A1* likely leading to a lack of collagen VII production in all three puppies is compatible with DEB, as this gene has multiple documented variants known to cause human DEB [1,2,3,4,5,6,7,8], and is found in two other canine cases of DEB [14,15,16].

The genome of one puppy was sequenced and assessed for homozygous and heterozygous genetic variants of candidate genes known to cause EB. We identified a homozygous variant of *COL7A1* in this puppy, and later confirmed that the other two were homozygous for the same variant, and that it was not seen in 12 control Basset Hounds nor in more than 600 dogs from other breeds. We were unable to obtain samples from the non-affected close relatives to confirm the co-segregation of the variant with the DEB phenotype in the family. The genotype distribution and the family history with non-affected parents and a close inbreeding loop strongly suggested autosomal recessive inheritance.

The identified variant involves a complex duplication event in an early exon that likely renders the gene inactive via a frameshift and resultant premature termination codon (PTC). Thus, we suspect that the collagen VII alpha 1 chain, the protein product of *COL7A1*, is not produced in affected puppies. While this specific variant appears to be unique to this family of puppies, a nonsense variant that resulted in PTC and absence of functional collagen VII alpha 1 chains caused similar lesions in Central Asian Shepherd dogs diagnosed with recessive DEB [14].

Autosomal recessive DEB is associated with PTC-producing variants of *COL7A1* in humans. Human *COL7A1* variants leading to PTCs were identified across multiple locations in the *COL7A1* gene in patients with autosomal recessive DEB (RDEB; OMIM #226600) [1,8]. Homozygous PTCs or compound heterozygous PTCs can lead to a lack of COL7A1 protein in humans, and the clinical picture includes severe blistering and erosions with onset at birth, pseudosindactyly, and contracted joints [8]. However, Varki et al. discuss instances in which an individual with PTCs on both alleles of *COL7A1* have a mild phenotype [8]. One such case was determined to be due to in-frame exon skipping through aberrant splicing [8,22]. In the investigated Basset Hound puppies, the presence of a homozygous frameshift and PTC with severe clinical signs and onset at birth are consistent with absence of a functional COL7A1 protein. Collagen VII is the main component of U-shaped structures called anchoring fibrils that help adhere the epidermis to the dermis by attaching to the lamina densa at both ends, with the middle extending into the dermis, entrapping interstitial collagen [23]. Variants that cause defects in, or in this case lack of collagen VII, compromise the stability of the epidermal–dermal junction, and it follows that areas subject to the most friction in nursing puppies—paw pads, esophagus, tongue, and the nasal planum—will be most affected, as seen in the puppies studied here [23].

Sub-classification of EB can utilize periodic acid–Schiff staining patterns and ultrastructural morphology of the basement membrane zone [4,9,24]. In this case, neither yielded definitive results—perhaps due to deposition of fibrin on the edges of advanced bullae (producing staining on both sides of the bullae) and post-mortem decomposition (sub-optimal preservation of membranes, filaments, and fibrils). When electron microscopy is used for diagnosis in humans, guidelines suggest that freshly induced blisters are sampled, as lesions more than 1 h old may produce ambiguous results [24].

Our study thus shows the value of a precision medicine approach and the molecular genetic characterization of dogs with suspected inherited disease. Whole genome sequencing is becoming increasingly efficient at identifying causative pathogenic variants, enabling precise diagnosis even when no fresh samples for a sophisticated histological investigation are available. Knowledge of the presumed causative genetic defect will enable genetic testing and the detection of healthy carriers to avoid the unintentional breeding of further affected dogs.

## 5. Conclusions

We found a homozygous complex duplication in *COL7A1* likely leading to the absence of functional COL7A1 protein in three neonatal, sibling Basset Hounds with severe and extensive blistering of the paw pads and nasal planum, nail loss, and tongue and esophageal involvement consistent with DEB. There are only a few documented cases of DEB in dogs, and to our knowledge, this is the first instance described in Basset Hounds.

## Figures and Tables

**Figure 1 genes-11-01458-f001:**
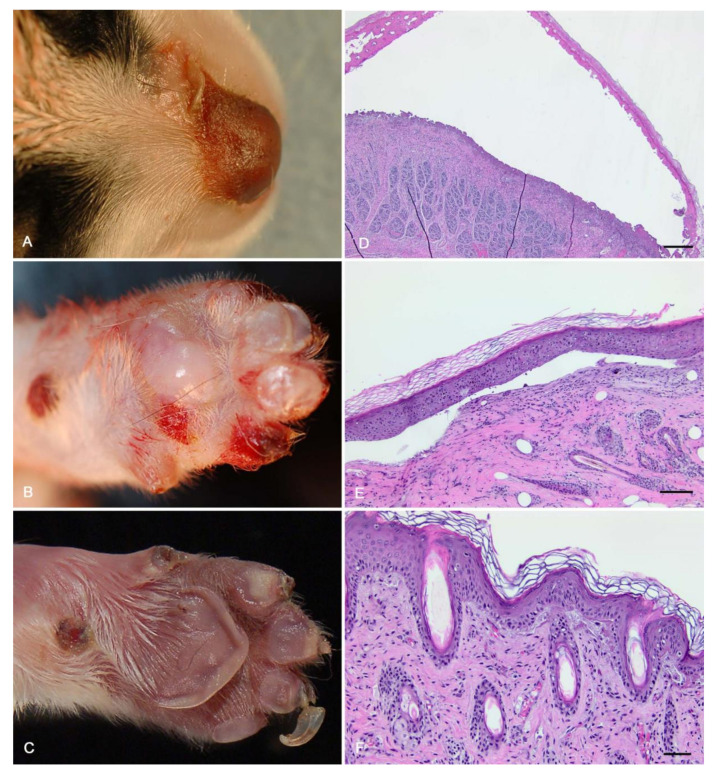
Gross necropsy with histopathology of three puppies with epidermolysis bullosa (EB). (**A**–**C**) All three dogs had bullae of the nasal planum and dorsal muzzle, and two had areas of ulceration. (**D**,**E**) Microscopic exam of the euthanized puppy found sub-basilar bullae. Scale bars = 500 μm. (**F**) In adjacent, less affected areas, there was subepidermal clefting with accumulation of proteinaceous fluid and a small amount of necrotic debris. There was also mild perivascular lymphoplasmacytic dermatitis. Scale bar = 50 μm.

**Figure 2 genes-11-01458-f002:**
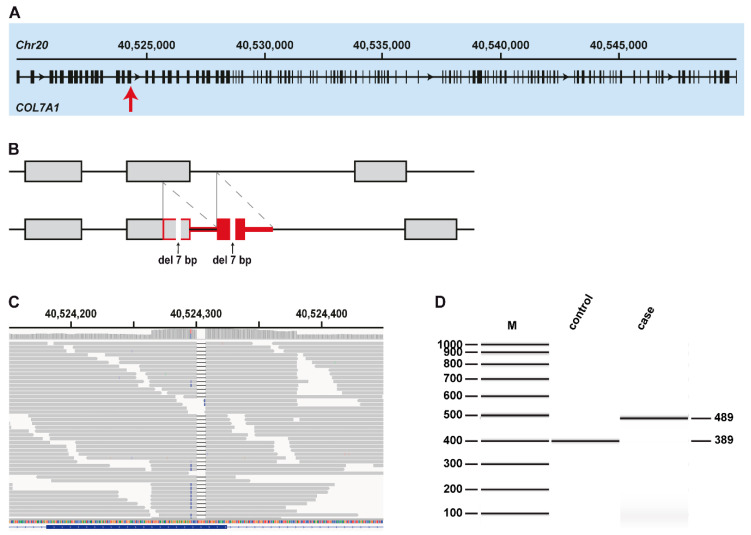
Details of the complex duplication in *COL7A1*. (**A**) Overview of the *COL7A1* gene with the intronic and exonic regions. The major transcript isoform (NM_001002980.1) is encoded by 118 exons. The position of the variant is indicated by an arrow. (**B**) Schematic illustration of the variant, where the exons are displayed with boxes and the introns with lines in between. The vertical grey lines indicate the position of the duplicated region, which is framed in red in the mutant allele at the bottom. The oblique dashed lines indicate the position of the actual duplication, which is colored solid red. The arrows indicate the deletion of seven nucleotides in both copies of the duplicated sequence. (**C**) Integrative Genomics Viewer (IGV) screenshot showing the short-read alignments of the dystrophic epidermolysis bullosa (DEB)-affected puppy at the position of the duplication. The increased coverage at the top is characteristic for a duplication. The deletion in both copies of the duplicated sequence is visible in the short-read alignments. (**D**) Fragment analyzer gel image with the PCR amplicons of a healthy control Basset Hound and a DEB-affected Basset Hound, indicating the insertion of 100 nucleotides in the mutant allele.

**Table 1 genes-11-01458-t001:** Results of variant filtering in an affected Basset Hound and 73 control genomes.

Filtering Step	Homozygous Variants	Heterozygous Variants
All variants	3,198,983	2,768,869
Private variants	18,820	80,057
Protein-changing private variants	49	174
Protein-changing private variants in 37 candidate genes	3	0

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
