# Peer review of "A COL7A1 Variant in a Litter of Neonatal Basset Hounds with Dystrophic Epidermolysis Bullosa"

_genes, 2020, doi:10.3390/genes11121458_

Round 1
Reviewer 1 Report
Dear authors,
This manuscript addresses DEB in Basset Hound 3 littermates and a putative candidate variant in COL7A1 was identified by WGS and comparison to unaffected dogs of the same breed. The identification of a major locus underlying this disease with deadly consequences for the affected puppies can lead to significant implications for animal welfare and health. Reading the manuscript was a joy, it is well written and easy to follow, the reported genetic analysis is sound and the results are relevant.
Therefore, I just have a few minor points/ comments.
All the best!
Minor:
L146: Please define “private variants”, e.g. variants that only occurred in the affected littermates
L189: Can you rewrite/ clarify “in a cohort of 15 Basset Hounds”? I assume the 15 dogs comprise of the 3 littermates and the 12 “controls”, but “a cohort” sounds a bit generic. Maybe “We confirmed the exact sequence of the variant in COL7A1 by Sanger sequencing (Figure S1) in one of the affected puppies and genotyped all three puppies and additional 12 control Basset Hounds for this variant”?
L196: Is there a word missing after “differential” (e.g. diagnosis)?
L204: It’s unfortunate that you were not able to get blood samples for the other littermates, but nonetheless, what is your hypothesis of mode of inheritance?
L234-237: The statement is very general, what are the specific implications for DEB? Would marker-assisted selection be possible?
Author Response
(1)
L164: Please define “private variants”, e.g. variants that only occurred in the affected littermates.
Response: We modified the sentence in line 162, so that this now includes a definition of private.
(2)
L189: Can you rewrite/ clarify “in a cohort of 15 Basset Hounds”? I assume the 15 dogs comprise of the 3 littermates and the 12 “controls”, but “a cohort” sounds a bit generic. Maybe “We confirmed the exact sequence of the variant in COL7A1 by Sanger sequencing (Figure S1) in one of the affected puppies and genotyped all three puppies and additional 12 control Basset Hounds for this variant”?
Response: Revised accordingly.
(3)
L196: Is there a word missing after “differential” (e.g. diagnosis)?
Response: Differential is sometimes used as an abbreviation for differential diagnosis. To avoid any misunderstanding, we added the word diagnosis to the text.
(4)
L204: It’s unfortunate that you were not able to get blood samples for the other littermates, but nonetheless, what is your hypothesis of mode of inheritance?
Response: We added the following sentence (lines 209-211): “The genotype distribution and the family history with non-affected parents and a close inbreeding loop strongly suggested autosomal recessive inheritance.”
(5)
L234-237: The statement is very general, what are the specific implications for DEB? Would marker-assisted selection be possible?
Response: We thank the reviewer for this important comment. We added a sentence stating that our findings enable genetic testing and the eradication of this disease (lines 243-245).
Reviewer 2 Report
The manuscript described the new mutation in COL7A1 gene in the dog breed Basset Hounds. The COL7A1 mutations were shown to induce dystrophic type of epidermolysis bullosa. Dogs and primates are the only species groups in which the development of EB was demonstrated. The mutation described represents an excellent example illustrating the advantages of study domestic dogs as a model system for inherited diseases. There are a few examples of DEB development in dogs and this the first describing the disease in Basset Hounds specifically.
In this research, three neonatal sibling Basset Hound out of 12 in the litter- were studied, which demonstrated the severe and extensive blistering of the paw pads and nasal planum, nail loss, and tongue and esophageal involvement. All puppies with devastating anomalies were proved to have the same mutation which represents large duplication with 7 nt deletion.
The manuscript describes the excellent done work both on the level of the whole genome sequencing part and analysis, and on the tissue analysis of expression abnormalities. The excellent experiments of ultrastructural skin investigation by electron microscopic investigation was shown in the Supplement, Fig S2.
Then allow me to add some critical remarks.
1.Authors write about DEB diagnosis in human, describing it using the old “Hallopeau-Siemens” variant of sybtype, meanwhile, as far as I'm concerned, the new classification is applied now for EB (and with RDEB severe generalized or localisata, particularly). We recommend to see the following review of C. Has et al., 2020 https://onlinelibrary.wiley.com/doi/10.1111/bjd.18921. Briefly, Epidermolysis Bullosa Dystrophica, AR, recessive DEB, OMIM 120120, MIM 226600. It seems to be also worthwhile to present the link to the OMIM data base (if this variant is not presented in the OMIA data base) in spite of the fact that it is human, but specifically deals with inherited diseases.
2.The author writes “The variant causes a frameshift, predicted to truncate 2260 amino acids (76%) from the C-terminus of the wildtype COL7A1 protein, NP_01002980.1:p.(Val677Serfs*11)”. It is unclear what resource did they have to make the prediction? I mean, that the link must be done, but it may be, that we simply could not manage to find it.
3.The region of the newly reported mutation is quite interesting because it belongs to the part of FN3 domains of COL7, whose form the repeats and is organized in a separate FN like domains structure. The specific properties of such domain organization are the regularity of exons open reading frames, namely one open reading frame for several following exons, which are ended with a full codon. If mutation takes place, and, particularly a disturbance of splicing takes place, it may cause the exon skipping without preliminary stop codon (PTC) generation, or it may induce PTC and subsequent degradation. The exon skipping process is reported for COL7A1 mutations. Moreover, the authors briefly mention this in the Discussion. In this case, due to the FN domain encoding exons structure this probability should be tested. It was reported that partly functional short mutant proteins produced as a result of exon skipping events. For example, as was demonstrated in this case report: https://pubmed.ncbi.nlm.nih.gov/26076072. By the way, this work described the area of just the same exon 15 of COL7A1. If the ORF of exons differs from this of COL7, the mutation could lead to accumulation of a non-desirable presumably toxic protein, without PTC and nonsense mediated decay of RNA.
Theoretically, the effect of this new EB inducing mutation could be reproduced in an artificial model system by the expression of this genetic variant in a heterological system. The recombinant protein technology even in the case of COL7 protein had been developed and is allowed to produce the protein which in such a case, is smaller than a human's. Although we do not think it is really necessary, the contemporary bioinformatics resources easily permit the exqusition of such types of data. For example, it is NETGENE2 service or Splice Site Prediction by Neural Network on https://fruitfly.org/seq_tools/splice.html (concerning human genome) of BDGP resource. Although such a comparison could be done for human genomes, there is no doubt that predictions will be exactly the same for canine genomes.
Minor comments: there are no scale bars on the Fig 1 (D, E, F). Figure 2D shows the fragments of PCR amplicons analyzed in agarose gel, it is better to show (may be in the Supplement) the uncut version of the image without data processing.
Author Response
(1)
Authors write about DEB diagnosis in human, describing it using the old “Hallopeau-Siemens” variant of sybtype, meanwhile, as far as I'm concerned, the new classification is applied now for EB (and with RDEB severe generalized or localisata, particularly). We recommend to see the following review of C. Has et al., 2020 https://onlinelibrary.wiley.com/doi/10.1111/bjd.18921. Briefly, Epidermolysis Bullosa Dystrophica, AR, recessive DEB, OMIM 120120, MIM 226600. It seems to be also worthwhile to present the link to the OMIM data base (if this variant is not presented in the OMIA data base) in spite of the fact that it is human, but specifically deals with inherited diseases.
Response: We briefly introduced the new nomenclature (Has et al. 2020) in the introduction, but accidentally slipped back to an old designation in the discussion. We now revised this according to the reviewer’s suggestion and removed the outdated term Hallopeau-Siemens DEB. We also added the link to the human OMIM entry for RDEB (line 220).
(2)
The author writes “The variant causes a frameshift, predicted to truncate 2260 amino acids (76%) from the C-terminus of the wildtype COL7A1 protein, NP_01002980.1:p.(Val677Serfs*11)”. It is unclear what resource did they have to make the prediction? I mean, that the link must be done, but it may be, that we simply could not manage to find it.
Response: The predicted translated protein was based on the assumption of unchanged splicing. We consider it unlikely that the partially duplicated exon 15 will be incorporated into the mutant transcript as this partially duplicated exon lacks a splice acceptor motif (more precisely: there is no 3’-splice site upstream of the duplicated partial exon). Therefore, in order to determine the translated protein, we used a hypothetical mutant COL7A1 cDNA sequence, which differs from the wildtype cDNA by lacking the 7 nucleotides deleted within exon 15 (c.2028_2034del).
We slightly expanded the results section and hope that it is now clear how we derived the variant designation on the protein level. We emphasized that this is a prediction based on the available evidence at the genomic DNA level (lines 173-174).
(4)
The region of the newly reported mutation is quite interesting because it belongs to the part of FN3 domains of COL7, whose form the repeats and is organized in a separate FN like domains structure. The specific properties of such domain organization are the regularity of exons open reading frames, namely one open reading frame for several following exons, which are ended with a full codon. If mutation takes place, and, particularly a disturbance of splicing takes place, it may cause the exon skipping without preliminary stop codon (PTC) generation, or it may induce PTC and subsequent degradation. The exon skipping process is reported for COL7A1 mutations. Moreover, the authors briefly mention this in the Discussion. In this case, due to the FN domain encoding exons structure this probability should be tested. It was reported that partly functional short mutant proteins produced as a result of exon skipping events. For example, as was demonstrated in this case report: https://pubmed.ncbi.nlm.nih.gov/26076072. By the way, this work described the area of just the same exon 15 of COL7A1. If the ORF of exons differs from this of COL7, the mutation could lead to accumulation of a non-desirable presumably toxic protein, without PTC and nonsense mediated decay of RNA.
Theoretically, the effect of this new EB inducing mutation could be reproduced in an artificial model system by the expression of this genetic variant in a heterological system. The recombinant protein technology even in the case of COL7 protein had been developed and is allowed to produce the protein which in such a case, is smaller than a human's. Although we do not think it is really necessary, the contemporary bioinformatics resources easily permit the exqusition of such types of data. For example, it is NETGENE2 service or Splice Site Prediction by Neural Network on https://fruitfly.org/seq_tools/splice.html (concerning human genome) of BDGP resource. Although such a comparison could be done for human genomes, there is no doubt that predictions will be exactly the same for canine genomes.
Response: Thank you very much for pointing out the interesting case report on the human patient with exon 15 variants. We think that the situation in this human patient is slightly different as this patient had a SNV affecting the first base of exon 15 and thus a variant directly affecting the splice acceptor site. The 7 bp deletion in the Basset Hounds is 121 nt downstream of the beginning of the exon and 16 nt upstream of the end of the exon. It is not very close to either the splice acceptor or the splice donor. Therefore, a prediction program that only analyzes the consensus sites at the exon/intron junctions (such as e.g. Splice Site Prediction) does not predict any changes of the splicing of the mutant exon 15 with the 7 bp deletion.
We ran Splice Site Prediction on the entire mutant allele including the partially duplicated exon and it still did not predict any new splicing sites compared to the wildtype allele. Therefore, the results of the bioinformatics prediction agree with our subjective evaluation that the most likely splicing pattern of the mutant allele will include exon 15 with the 7 nt deletion spliced to exon 16 (without inclusion of any of the duplicated sequence). This will lead to a frameshift and an early premature termination codon (PTC) and is consistent with the observed severe clinical phenotype in the dogs.
We greatly regret that no suitable samples for experimental verification of these hypotheses are available (fresh skin tissue samples for RNA isolation).
Minor comments
(4)
There are no scale bars on the Fig 1 (D, E, F).
Response: We added the scale bars as requested.
(5)
Figure 2D shows the fragments of PCR amplicons analyzed in agarose gel, it is better to show (may be in the Supplement) the uncut version of the image without data processing.
Response: We did not run an agarose gel but performed the fragment size analysis on a capillary gel electrophoresis instrument (5200 Fragment Analyzer). While capillary gel electrophoresis is more sensitive and provides a higher resolution than conventional agarose gel electrophoresis, it is unfortunately not possible to give “an image without data processing”. We added the used methodology to section 2.7 (lines 121-122).
Additional change: We slightly revised the discussion to make a clearer distinction between the COL7A1 gene and the COL7A1 protein.